# Optimized Replication of Arrayed Bacterial Mutant Libraries Increases Access to Biological Resources

Julia L. E. Willett,[a] Aaron M. T. Barnes,[a*] Debra N. Brunson,[b] Alexandre Lecomte,[c] Ethan B. Robertson,[a] Gary M. Dunny[a]

[a]Department of Microbiology and Immunology, University of Minnesota Medical School, Minneapolis, Minnesota, USA
[b]Department of Oral Biology, University of Florida College of Dentistry, Gainesville, Florida, USA
[c]Université Paris-Saclay, INRAE, AgroParisTech, Micalis Institute, Jouy-en-Josas, France

Julia L. E. Willett and Aaron M. T. Barnes contributed equally to this work. The co-first author designation was decided by reverse alphabetical order.
Debra N. Brunson and Alexandre Lecomte contributed equally to this work.

**ABSTRACT** Biological collections, including arrayed libraries of single transposon (Tn) or deletion mutants, greatly accelerate the pace of bacterial genetic research. Despite the importance of these resources, few protocols exist for the replication and distribution of these materials. Here, we describe a protocol for creating multiple replicates of an arrayed bacterial Tn library consisting of approximately 6,800 mutants in 96-well plates (73 plates). Our protocol provides multiple checkpoints to guard against contamination and minimize genetic drift caused by freeze/thaw cycles. This approach can also be scaled for arrayed culture collections of various sizes. Overall, this protocol is a valuable resource for other researchers considering the construction and distribution of arrayed culture collection resources for the benefit of the greater scientific community.

**IMPORTANCE** Arrayed mutant collections drive robust genetic screens, but few protocols exist for replication of these resources and subsequent quality control. Increasing the distribution of arrayed biological collections will increase the accessibility and use of these resources. Developing standardized techniques for replication of these resources is essential for ensuring their quality and usefulness to the scientific community.

**KEYWORDS** *Enterococcus faecalis*, biological collections, functional genomics, genetic tools, scientific rigor, transposon mutagenesis

Mutagenesis of a given organism, followed by phenotypic selection or measurement of mutant fitness, is a cornerstone of experimental microbial genetics. High-quality, publicly available collections of mutants, such as the *Escherichia coli* Keio collection (1), *Bacillus subtilis* single-gene knockout libraries (2), and the *Staphylococcus aureus* USA300_FPR3757 transposon (Tn) mutant library (3), greatly accelerate the pace at which research can be performed and enhance scientific rigor and reproducibility. Construction of such resources requires significant time, labor, and resources, and it is inefficient for multiple laboratories to generate redundant biological resources. The research community would benefit from increased generation, replication, and dissemination of resources such as arrayed bacterial mutant libraries. Although robust protocols have been developed for manual or robotic arraying of colonies and mapping of arrayed collections of mutants (4–8), few exist for the replication of arrayed culture collections (9–11). Here, our goal was to establish a protocol and collection of best practices to minimize contamination and genetic drift of arrayed bacterial culture collections while increasing accessibility for other researchers.

We previously described the generation and application of an arrayed library of Tn mutants in *Enterococcus faecalis* OG1RF, consisting of ~15,000 individual clones (12). From this library, two targeted sequence-defined *mariner* Tn (SmarT) libraries were generated (5). The first SmarT library consists of 6,829 Tn mutants arrayed across 96-well plates (*n* = 73 plates),

Address correspondence to Julia L. E. Willett, jwillett@umn.edu.

*Present address: Aaron M. T. Barnes, Minnesota Department of Health, St. Paul, Minnesota, USA.

The authors declare no conflict of interest.

with insertions in approximately 70% of annotated genes and intergenic regions in OG1RF. The second SmarT library consists of 1,946 Tn insertions in poorly characterized genes and intergenic regions and was designed to facilitate genetic screens targeting uncharacterized regions of the genome (13). Both libraries are also available in pooled formats and have been used extensively to identify *E. faecalis* genes required for biofilm formation, metabolism, responses to antibiotics, phage infection, vaginal colonization in a mouse model, and polymicrobial interactions involving *E. faecalis* (14–21).

In addition to these genetic screens, hundreds of individual Tn mutants have been distributed to domestic and international laboratories. We regularly receive requests for the entire Tn library, but it is not feasible to generate a new copy of the entire arrayed library for each individual request. Therefore, we sought to perform a large-scale replication of the larger SmarT library (6,829 mutants) to increase the accessibility of this resource for other laboratories, ensure quality control of the collection, and avoid genetic drift by decreasing the number of freeze/thaw cycles for the original library plates. Here, we present a protocol for efficient manual replication of arrayed library resources, including estimation of the time required (person hours). This protocol does not require access to robotic handling systems, making it feasible for researchers who do not have access to this specialized equipment. This protocol can be scaled to accommodate libraries of different sizes, as well as different numbers of replicates. We also describe multiple quality control checks throughout the process and compare sequencing-based verification of pooled mutants with previously published results. Additionally, because of ongoing supply chain difficulties due to the coronavirus disease 2019 (COVID-19) pandemic, we consider multiple options for consumables required throughout, as well as ergonomic considerations for technical staff.

## RESULTS

**Abbreviated protocol for large-scale replication and quality control of arrayed Tn resources.** We manually created 15 copies of the 73-plate SmarT library over a 2-week period in April 2022. Required supplies and consumables are listed in Table 1. The approximate timeline for library replication on this scale is outlined in Tables 2 and 3. Additional protocol details can be found in File S1 in the supplemental material. To avoid repeated freeze/thaw cycles of the original library plates, all copies were created at the same time. An overview of the process is shown in Fig. 1. Frozen SmarT library stock plates were used to inoculate deep 96-well plates containing brain heart infusion (BHI) broth. Cultures were grown overnight and manually inspected for contamination of known blanks or lack of growth. If plates had either contaminated wells or wells with no growth, then the entire plate was discarded, and a new deep-well plate was inoculated. Overnight cultures were transferred from deep-well plates to prelabeled 96-well plates containing glycerol to generate individual library sets, which were stored at −80°C.

Multiple quality control checkpoints were used to prevent contamination. The entire process was carried out in a class 2 (A2) biological safety cabinet using biosafety level 2 (BSL-2) practices. Deep-well plates were filled with BHI broth and incubated overnight at 37°C prior to inoculation to ensure a lack of contamination. Replicating pins used for inoculation from freezer stock plates were sterilized in a series of ethanol baths between plates. To ensure against dilution effects that could reduce the effectiveness of alcohol-based disinfection, the ethanol baths were completely replaced after inoculation of four deep-well plates. Replicating pins were also autoclaved every night.

The original SmarT library layout included a mapped series of known blank/empty wells in each plate (5). These wells were manually inspected in the deep-well plates, and any plates with contamination were discarded and reinoculated. Any deep-well plates in which mutants did not grow as expected were also discarded and reinoculated. Because the deep-well plates do not fit in standard plate readers, optical density (OD)/absorbance was not measured. We recognize this as a limitation of the protocol, because such data could be important for identifying mutants with reduced overnight growth that is not detectable by eye. We strongly recommend that users who adapt this protocol for their own biological collections measure endpoint $OD_{600}$ values and provide these as a data set for the community.

**TABLE 1** Required reagents, consumables, and equipment

| Item | Amount required | Notes |
|---|---|---|
| 96-well plates (73 per set) | 1,095 plates for 15 copies, 1,200 plates to ensure sufficient backups, 1,300 plates to ensure sufficient quantities for measuring endpoint $OD_{600}$ values after growth in deep-well plates | Plates should be sterile. Sterile plates can typically be purchased in multipacks (10 or 20 per sleeve) but, due to ongoing supply chain disruptions during the pandemic, we were able to purchase only individually wrapped plates. We do not recommend purchasing individually wrapped plates, due to the time required to unwrap each plate. |
| 96-well plate labels | 1,095 plates for 15 copies, 1,200 plates to ensure sufficient backups | Ensure that the labels are rated for cryogenic preservation. Labels can be generated and printed in sheets and then applied to individual plates. |
| Glycerol | 10.5 L of a 50% (v/v) solution for 15 copies; the recommended volume is 12 L | Glycerol should be prepared in advance as a 50% (v/v) solution in 100-mL bottles. |
| Growth medium (BHI broth) | 12.6 L required for 15 copies (1.8 mL/well × 73 deep-well plates); the recommended volume is 14 L | BHI broth should be prepared in advance in 100-mL bottles. |
| Replicating pins | >1 | Metal pins that can be sterilized and autoclaved are recommended; pins should be autoclaved every day. |
| Ethanol | >4 L | The ethanol is used to sterilize replicating pins between plate inoculations. Ethanol baths (made by filling empty pipette tip boxes) should be changed every 4 plates to ensure sterility. |
| Electronic P1000 pipette | >1 | We do not recommend performing pipetting and plate replication on this scale with manual pipettes due to the possibility of repetitive motion injuries. Even with electronic pipettes, we recommend alternating users or limiting the number of plates prepared per day for ergonomic considerations. |
| P1000 pipette tips | 7 boxes for aliquoting BHI broth into deep-well plates | Pipette 1.8 mL in 900-$\mu$L increments and change pipette tips after each deep-well plate. |
| Electronic P200 pipette | >1 | We do not recommend performing pipetting and plate replication on this scale with manual pipettes due to the possibility of repetitive motion injuries. Even with electronic pipettes, we recommend alternating users or limiting the number of plates prepared per day for ergonomic considerations. |
| P200 pipette tips | 1 box per 96-well plate in the original library (73 in the SmarT library) | Prepare and set aside additional boxes in case they are needed. |
| Foil seals | 1 seal per new 96-well plate (1,095 for 15 copies of the SmarT library) plus replacements for the original library (73 for the original SmarT library) | Ensure that the seals are rated for cryopreservation. |
| Adhesive seal roller | >1 | Small foam or rubber rollers (3–4 inches) can be obtained from craft stores (sold as wallpaper or printmaking rollers) and scientific supply companies. |

Importantly, we did not find any mutants in the entire 6,829-clone library that have lost viability since the initial library construction.

New copies of library plates were prefilled with glycerol and capped with sterile foil seals after addition of bacterial cultures. Plates were mixed by inversion after sealing (instead of pipetting) due to the number of samples. To ensure that this would not introduce cross-well contamination, we first empirically investigated the potential for cross-contamination. We filled a 96-well plate with glycerol, buffer, and bromophenol blue for visualization. This plate was covered with a foil seal, vortex-mixed at 2,500 rpm for 5 min, and incubated overnight on a rotating shaker. This agitation exceeded the brief mixing process performed with Tn library plates containing cultures and glycerol. No dye leakage or damage to the foil seal was observed (Fig. 2A), suggesting that brief mixing of Tn library plates would not cause intraplate contamination. To quantitatively assess whether the exact process used with library plates would introduce interwell contamination, we filled a 96-well plate with *E. faecalis* OG1RF ($\sim$$10^7$ CFU/mL) or medium blanks. The plate was sealed, inverted, and incubated at 37°C overnight. Any well-to-well leakage would result in growth of OG1RF in the medium blank wells. We did not observe any contamination or growth outside the wells that were inoculated (Fig. 2B), strengthening our conclusion that the library plate mixing process does not cause intraplate contamination. This is consistent with our experience creating and handling the original arrayed Tn library.

**TABLE 2** Timeline for generation of 15 copies of a 73-plate library

| Task | Time | Notes |
|---|---|---|
| Ordering materials | Variable | It is recommended to order ~10% extra of each reagent/supply. |
| Generating plate labels | ~4 h | Ensure that cryogenic labels are used. |
| Preparing autoclaved reagents (BHI broth and glycerol) | ~8 h | Prepare in 100-mL or 250-mL bottles to decrease the chances of contamination by repeated pipetting. |
| Labeling and sorting 96-well plates | ~30 person h for 1,200 plates | |
| Distributing BHI broth into deep-well plates | ~35 person h | Plus overnight incubation (performed prior to inoculation to ensure a lack of contamination). |
| Distributing 50% glycerol into 96-well plates | ~35 person h | |
| Inoculating 96-well plates | ~1 person h per 4 plates | Work in sets of 4 plates. Plates need ~10 min to thaw enough to use inoculating pins. This step requires overnight incubation. |
| Checking inoculated plates for blank wells and contamination | ~1 person h per set of 16 plates (2 people × 0.5 h) | |
| Distributing cells from deep-well plates to new library copies and sealing the plates with foil | ~1 person h per deep-well plate (15 new 96-well plates) | |
| Measuring endpoint $OD_{600}$ values from mutants grown in deep-well plates | ~1 person h per day | |
| Sorting and storing new library plates at −80°C | ~2 person h per day | |

**Pooling of the SmarT library from 96-well plates and Tn sequencing.** The SmarT libraries are available in arrayed format and as a pooled version, in which all Tn mutants have been combined in equal amounts to facilitate Tn sequencing (Tn-Seq) or similar genetic screens. Previously, the pooled versions were created by plating aliquots of each strain on BHI agar plates, followed by scraping and combining all mutants (5). This was done to ensure that roughly the same numbers of mutants would be present in the pooled library regardless of *in vitro* growth defects in liquid medium and to screen each mutant stock for contamination. To determine whether we could pool liquid cultures of the SmarT library after growth in deep-well plates and still achieve a similar balance of mutants, we combined ~200 $\mu$L remaining from each deep-well plate after distribution of the cultures into the new library plates. DNA was extracted from the pooled cultures, and Tn abundance was determined using Illumina sequencing and previously established protocols (5). These results were compared to those for samples extracted from the original pooled SmarT library in a previous experiment (17).

We first compared the number of Tn mutants identified from sequencing with the known number of mutants in the arrayed library ($n = 6,829$). In the new pooled library, 250 mutants were missing (0 reads) in all extracted replicates (358, 336, and 347 mutants in the individual replicates) (see Table S1 in the supplemental material). A total of 192 mutants had 0 reads in all previously sequenced samples (275, 259, 268, and 267 mutants in the individual replicates) (17). A total of 166 mutants had 0 reads mapped in all replicates of the old and new pooled libraries. These mutants might be missing due to incomplete lysis of cells (perhaps due to physiological changes due to the disrupted gene), instability of the Tn

**TABLE 3** Planned versus actual timeline for replication of original plates in SmarT library

| | No. of plates | | | | | |
|---|---|---|---|---|---|---|
| | Planned | | | Actual | | |
| Day | Transferred | Inoculated | Total completed | Transferred | Inoculated | Total completed |
| 1 | 0 | 10 | 0 | 0 | 10 | 0 |
| 2 | 8 | 16 | 8 | 8 | 16 | 8 |
| 3 | 14 | 16 | 22 | 8 | 16 | 16 |
| 4 | 14 | 16 | 36 | 10 | 22 | 26 |
| 5 | 14 | 0 | 50 | 15 | 0 | 41 |
| 6 | 0 | 16 | 50 | 0 | 16 | 41 |
| 7 | 14 | Remainder | 64 | 16 | 17 | 57 |
| 8 | Remainder | 0 | 73 | 14 | 0 | 70 |
| 9 | 0 | 0 | 0 | 0 | 3 | 70 |
| 10 | 0 | 0 | 0 | 3 | 0 | 73 |

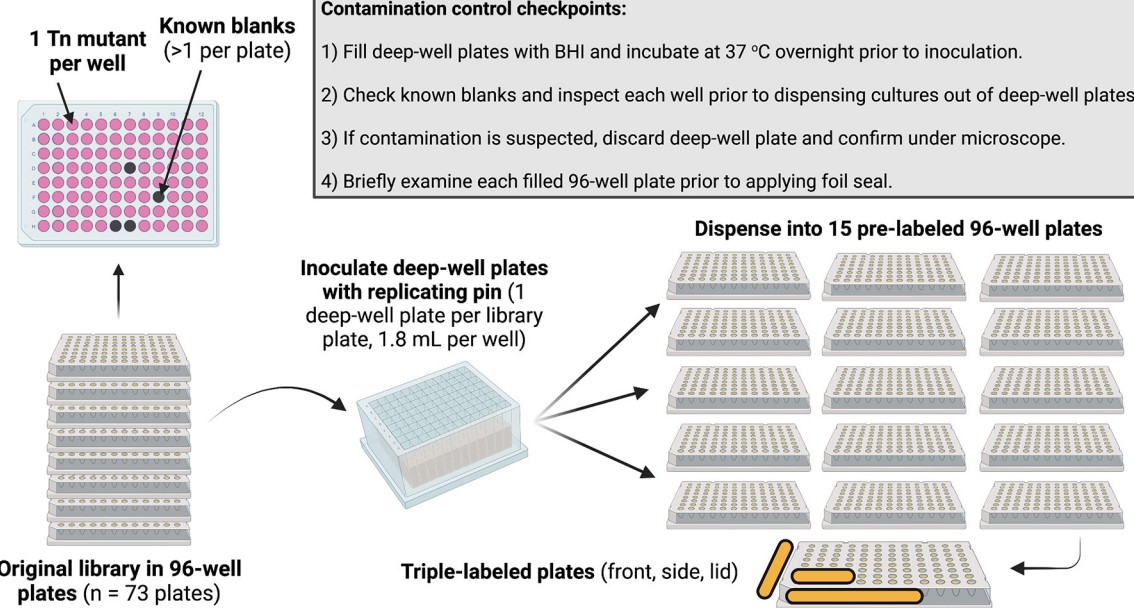

**FIG 1** Overview of arrayed library replication and quality control checkpoints. SmarT library plates were used to inoculate deep-well plates. Cultures were grown, examined for contamination, and then dispensed into 15 individual prelabeled, preloaded plates for new library sets. This image was created using BioRender.

insertion in the chromosome, or loss of DNA during preparation and sequencing. Because we did not find any mutants that did not grow in deep-well plates during library replication, we do not think that these mutants are missing from the sequencing results due to a loss of viability. We next examined the similarity in relative abundances of mutants in the original pooled library and the new library pooled from liquid cultures (Fig. 3A). In addition to a greater number of mutants with 0 reads, the new pooled library had a broader distribution of relative abundance frequencies (Fig. 3B). Low-abundance mutants in the new pooled library (relative abundance, 0 to 0.00001) had higher relative abundances in the original input library (Fig. 3C). However, mutants that were absent from or had relatively low abundances in the original pooled library also had low abundances in the new pooled library (Fig. 3D). Overall, we conclude that the original pooled library created by collecting cells grown on agar plates had a more even distribution of mutants than the new pooled library created by combining liquid cultures. Based on this, we strongly recommend generating new pooled libraries using our original method of plating and scraping individual mutants.

## DISCUSSION

Culture collections and arrayed mutant libraries are valuable biological resources that increase the throughput, rigor, and reproducibility of experiments across an entire scientific field. To avoid redundancy and wasted resources, these collections and libraries should be broadly available to researchers. While core facilities or private companies may have resources to generate arrayed library copies using robotic arraying and liquid-dispensing equipment, this remains beyond the reach of most academic research laboratories at many institutions. Therefore, we sought to establish a protocol for manual replication of arrayed library collections that would increase the accessibility of these biological resources while maintaining high quality control standards and preventing genetic drift due to repeated freeze/thaw cycles of arrayed culture collections.

Using this approach, we created 15 copies of a large arrayed *E. faecalis* OG1RF Tn library; we have already distributed most of these library sets to other research groups. We also pooled Tn mutants grown during library replication and used Tn-Seq to compare mutant distribution from this pool to the results for a previously generated pooled library in which individual mutants were scraped from agar plates. Although we found that a majority of Tn mutants (>96%) were present in our new pooled library, we observed greater representation

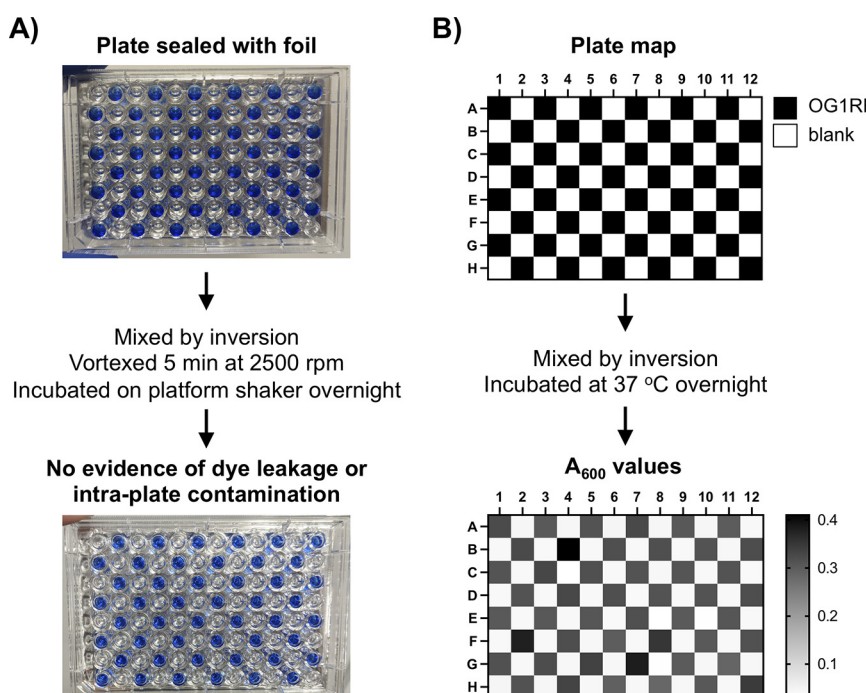

**FIG 2** Evidence that the sealing and mixing process does not introduce intraplate contamination. (A) A test plate with glycerol and bromophenol blue was sealed with foil, mixed by inversion, vortex-mixed, and incubated overnight on a shaking platform to ensure that the inversion process used to mix glycerol and bacterial cultures would not create contamination between wells. (B) A 96-well plate was inoculated with alternating wells of *E. faecalis* OG1RF ($\sim 10^7$ CFU/mL) or medium blanks. To mimic library plate processing, the plate was capped with a foil seal and mixed by inversion. The plate was incubated at 37°C overnight, after which an $OD_{600}$ measurement was taken to evaluate growth. The heatmap values are the averages of three independent biological replicates.

of low-abundance mutants using the previous approach of scraping and pooling mutants from agar plates. The liquid pooling approach described here could have been strengthened by measuring endpoint $OD_{600}$ values for each library plate after overnight growth and adjusting the volume pooled for each mutant accordingly. Although the plating and scraping approach is more labor-intensive, we strongly recommend using that approach to generate new pooled libraries, instead of pooling together liquid cultures. This methodology can guide the creation and distribution of arrayed mutant collections in a variety of microorganisms.

## MATERIALS AND METHODS

**Bacterial strains and culture conditions.** The 6,829-clone *E. faecalis* arrayed Tn library was previously generated and stored at −80°C (5, 12). BHI broth (BD Difco) was used for overnight growth. Prior to inoculation, plates were filled with BHI broth, incubated at 37°C, and manually inspected for contamination prior to inoculation. Tn mutants were inoculated, using a metal replicating pin (Boekel Scientific), into 2-mL deep 96-well plates (Biotix) containing 1.8 mL BHI broth. Plates were grown overnight at 37°C without shaking and were manually inspected for contamination prior to distribution of cultures to new library plates.

**Preparation of library plates.** Sterile flat-bottomed 96-well plates (Thermo Fisher Scientific) were labeled on the lid and two sides of the plate with printed cryovial labels (LabTAG). Labels contained the library copy number (from 1 to 15) and plate number (from 1 to 73). One hundred microliters of autoclaved 50% glycerol (VWR) was dispensed into each well using multichannel electronic pipettes.

**Generation of new library copies.** From each deep-well plate, 100 $\mu$L of overnight culture was distributed into 15 prelabeled 96-well plates (prefilled with 100 $\mu$L 50% glycerol) using multichannel electronic pipettes. Sterile AlumaSeal adhesive foil seals (Life Science Co.) were applied with a small rubber roller (Speedball). Plates were sorted and stored at −80°C.

**Evaluation of intraplate leakage.** A 96-well plate with 0.1% bromophenol blue in phosphate-buffered saline (pH 8) and 10% glycerol was set up in a checkerboard pattern, with a total volume of 200 $\mu$L per well. The plate was capped with an AlumaSeal adhesive foil seal (Life Science Co.) and sealed with a small rubber roller (Speedball). The plate was mixed by inversion, vortex-mixed for 5 min at 2,500 rpm on a multitube vortex-mixer (Benchmark Scientific), and incubated overnight on a Belly Dancer orbital shaker at $\sim$50 rpm. Leakage of dye between wells was evaluated by eye. For quantitative measurement of intraplate leakage, a 96-well plate was inoculated with alternating wells of *E. faecalis* OG1RF ($\sim 10^7$ CFU/mL) in BHI broth plus 10% glycerol or medium blanks. The plate was capped with a foil seal, mixed by inversion, and incubated at 37°C overnight, after

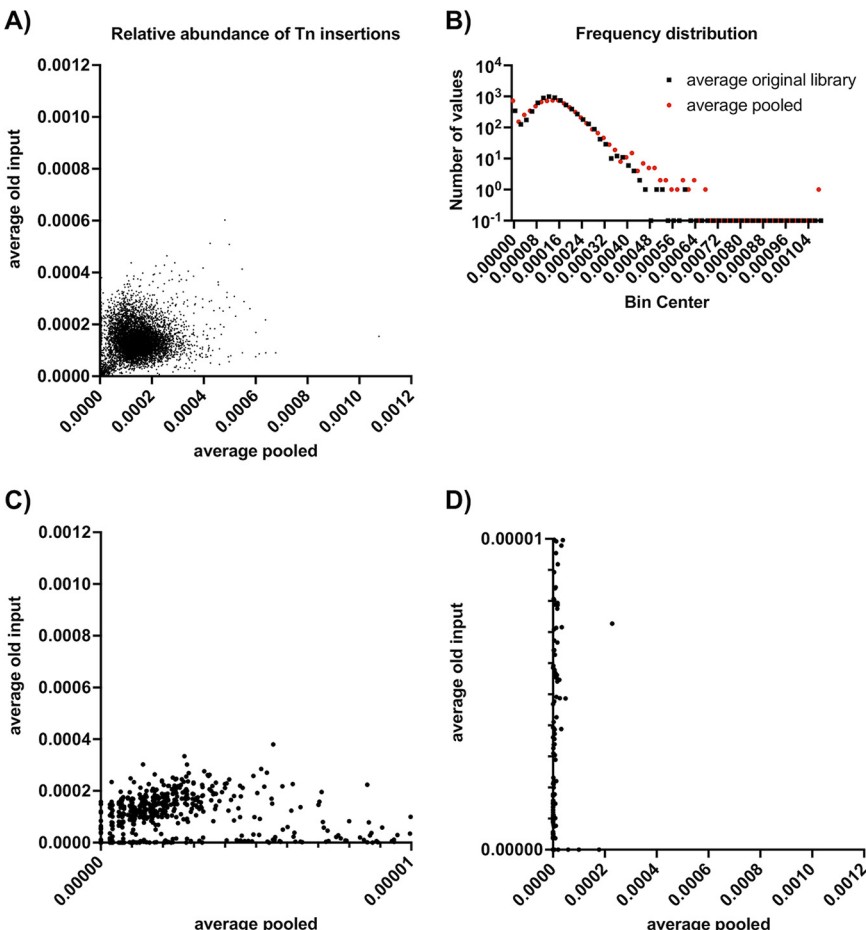

**FIG 3** Comparison of the SmarT library pooled from liquid cultures and that from the original pooled library format. (A) Relative abundances of Tn mutants in each library. (B) Distribution of relative abundances. (C) Abundances of low-abundance mutants from the new pooled library, relative to abundances in the original library. (D) Abundances of low-abundance mutants from the original library, relative to abundances in the new pooled library.

which an $OD_{600}$ measurement was taken using a Biotek Synergy H1 plate reader. Data shown are the average of three independent biological replicates.

**Preparation of pooled Tn samples for Tn-Seq and Tn-Seq analysis.** Two hundred microliters was pooled from each well of each deep-well plate after the distribution of mutants to new library plates. Samples were pooled in 50-mL Falcon tubes, pelleted at 6,000 rpm for 10 min in a Beckman Coulter Avanti JXN-30 floor centrifuge, and stored at −20°C until further use. DNA was extracted using a Qiagen DNeasy blood and tissue kit with a lysozyme pretreatment step, as described previously (17), and submitted to the University of Minnesota Genomics Center. Libraries were prepared using a NEBNext Ultra II FS DNA library preparation kit for Illumina and a Nextera XT Index kit v2 set A. Libraries were sequenced using a NextSeq P1 flow cell (150-bp paired-end reads), with ~2.1 million reads per sample (~300 reads/mutant). Mutant abundances were quantified using custom scripts, as described previously (5, 17).

**Data availability.** Tn-Seq data have been deposited in the NCBI GEO database under accession number GSE233193.

## SUPPLEMENTAL MATERIAL

Supplemental material is available online only.
**SUPPLEMENTAL FILE 1**, DOCX file, 0.1 MB.
**SUPPLEMENTAL FILE 2**, XLSX file, 1.3 MB.

## ACKNOWLEDGMENTS

This work was supported by a grant from Boehringer Ingelheim Fonds to A.L., American Heart Association fellowship 907592 to D.N.B., National Institutes of Health grant K99AI151080 to J.L.E.W., and National Institutes of Health grant R01AI122742 to G.M.D. A.L. was supported by a fellowship from Healthy as part of the France 2030 program ANR-11-IDEX-0003, from

the OI HEALTHI of the University Paris-Saclay and from the Mica division of INRAE of Jouy-en Josas.

We thank Cristel Archambaud and Pascale Serror for supporting the participation of A.L. and Jose Lemos for supporting the participation of D.N.B. We are also grateful to Pascale Serror and Jose Lemos for helpful comments on the manuscript.

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
