## [Reviewer comments · Microbiology Spectrum]

Microbiology Spectrum

Optimized replication of arrayed bacterial mutant libraries increase access to biological resources

Julia Willett, Aaron Barnes, Debra Brunson, Alexandre Lecomte, Ethan Robertson, and Gary Dunny

Corresponding Author(s): Julia Willett, University of Minnesota Twin Cities

Review Timeline:

Submission Date:	April 25, 2023
Editorial Decision:	May 19, 2023
Revision Received:	June 16, 2023
Accepted:	June 19, 2023

Editor: Justin Kaspar

Reviewer(s): The reviewers have opted to remain anonymous.

Transaction Report:

DOI: <https://doi.org/10.1128/spectrum.01693-23>

May 19, 2023

Dr. Julia Willett
University of Minnesota Twin Cities
Minneapolis, MN 55455

Re: Spectrum01693-23 (Optimized replication of arrayed bacterial mutant libraries increase access to biological resources)

Dear Dr. Julia Willett:

The referenced manuscript has been reviewed by two reviewers. Both have found the manuscript well written, with only a couple suggestions on potential improvements prior to publication. First, sequencing read data from Figure 3 should be available in a public repository. Second, reviewer 2 has provided two instances where quantitative quality control checkpoints could be added to the protocol. Please address these comments in a revision of the manuscript.

Link Not Available

Sincerely,

Justin Kaspar

Journals Department
Reviewer comments:

Reviewer #1 (Comments for the Author):

In this manuscript, Willett and colleagues describe a protocol for generating an arrayed Tn library in *E. faecalis* that is generally applicable to other bacterial species. Arrayed Tn libraries are important resources for conducting gene function screens in bacteria, but they are time consuming and labor intensive to generate. As the authors also mention, there are no clear guidelines on how to generate such resources, with the 'secrets' of how to generate these libraries arising through a combination of intuition, luck and working in labs with long histories of making these resources. In this regard, it is helpful to have an established, detailed, protocol for creating an arrayed mutant library. Although the procedure is written for replication of the *E. faecalis* SmarT library it could be applied to other bacterial species with modifications (e.g. growth media). The method is

rigorous and easy to follow, with the tables (1-3) being particularly helpful for following the procedure. Description of the reagents and equipment required is very detailed, and this will make the protocol easier to replicate. The goal of the manuscript was to provide a protocol for generation/replication of arrayed mutant libraries and the authors have successfully achieved this. I have no major or minor comments.

Reviewer #2 (Comments for the Author):

In this manuscript, Willett et al. develop a standardized protocol for the replication of strain libraries and apply it to copying a previously-created, transposon-mutagenized strain library of *Enterococcus faecalis* OG1RF. The authors use this protocol to create 15 copies of said library and discuss several quality control checkpoints, while exploring alternative strategies to pooled strain library creation. Given the increasing prevalence of high-throughput experiments within the field of microbiology, studies like this are needed and will proliferate concepts of industrial quality engineering. While the manuscript is well written and the experiments are straightforward, the reliance on qualitative observation for several of the quality control steps weakens the approach. The authors should implement a few simple changes to the protocol to improve the strength of the manuscript.

Major issues:

I agree with where the authors have placed quality control checkpoints within the library replication protocol but believe they could be made more rigorous by making these checks quantitative in nature.

1) Lines 120-125. The authors claim that because the mutant library was grown in deep well plates, and because deep well plates do not fit within standard plate readers, they did not measure optical density of overnight cultures. While the scale of the replication experiments is quite large, I find it difficult to believe that simply transferring a few hundred microliters of overnight culture to a standard 96-well plate and measuring OD600 would have been prohibitively difficult. The inclusion of an endpoint OD600 plate read for all library plates within the protocol would greatly improve the strength of the manuscript. By including this step, the authors would be able to better identify slow-growing contamination in blank wells and would have quantitative information on each mutant's endpoint growth --- valuable data to labs receiving the library and also data which could vastly improve liquid culture library pooling efforts.

2) Lines 130-137. My criticism in point 1 holds for the dye leakage experiments. Similar, quantitative experiments using dyes with well-characterized absorbance spectra and plate readers are commonplace in liquid handling quality engineering contexts. While I don't doubt the results of their experiment (shown in Figure 2), the manuscript could be greatly improved by using quantitative data to demonstrate the lack of well-to-well leakage. This could be visualized as a heatmap.

Minor issues:

The results of the liquid pooling experiment shown in Figure 3 are not surprising. The dropout or near drop out of certain strains is likely exacerbated by unequal cellular contributions from each of the overnight cultures, with slow growing mutants being added in lower quantities than fast or typical growing mutants. The authors should explicitly state that they recommend the original pooling protocol while providing advice for how the liquid pooling strategy could've been improved (endpoint OD600 measurements for all plates coupled with liquid contributions normalized to the individual mutant's optical density). I acknowledge that this last point would be incredibly tedious to accomplish manually, but inexpensive liquid handling robots are rapidly becoming available and accessible for many labs.

Staff Comments:

Preparing Revision Guidelines

For complete guidelines on revision requirements, please see the journal Submission and Review Process requirements at

<https://journals.asm.org/journal/Spectrum/submission-review-process>. **Submissions of a paper that does not conform to Microbiology Spectrum guidelines will delay acceptance of your manuscript. "**

Please return the manuscript within 60 days; if you cannot complete the modification within this time period, please contact me. If you do not wish to modify the manuscript and prefer to submit it to another journal, please notify me of your decision immediately so that the manuscript may be formally withdrawn from consideration by Microbiology Spectrum.

In this manuscript, Willett et al. develop a standardized protocol for the replication of strain libraries and apply it to copying a previously-created, transposon-mutagenized strain library of *Enterococcus faecalis* OG1RF. The authors use this protocol to create 15 copies of said library and discuss several quality control checkpoints, while exploring alternative strategies to pooled strain library creation. Given the increasing prevalence of high-throughput experiments within the field of microbiology, studies like this are needed and will proliferate concepts of industrial quality engineering. While the manuscript is well written and the experiments are straightforward, the reliance on qualitative observation for several of the quality control steps weakens the approach. The authors should implement a few simple changes to the protocol to improve the strength of the manuscript.

Major issues:

I agree with where the authors have placed quality control checkpoints within the library replication protocol but believe they could be made more rigorous by making these checks quantitative in nature.

- 1) Lines 120-125. The authors claim that because the mutant library was grown in deep well plates, and because deep well plates do not fit within standard plate readers, they did not measure optical density of overnight cultures. While the scale of the replication experiments is quite large, I find it difficult to believe that simply transferring a few hundred microliters of overnight culture to a standard 96-well plate and measuring OD600 would have been prohibitively difficult. The inclusion of an endpoint OD600 plate read for all library plates within the protocol would greatly improve the strength of the manuscript. By including this step, the authors would be able to better identify slow-growing contamination in blank wells and would have quantitative information on each mutant's endpoint growth --- valuable data to labs receiving the library and also data which could vastly improve liquid culture library pooling efforts.
- 2) Lines 130-137. My criticism in point 1 holds for the dye leakage experiments. Similar, quantitative experiments using dyes with well-characterized absorbance spectra and plate readers are commonplace in liquid handling quality engineering contexts. While I don't doubt the results of their experiment (shown in Figure 2), the manuscript could be greatly improved by using quantitative data to demonstrate the lack of well-to-well leakage. This could be visualized as a heatmap.

Minor issues:

The results of the liquid pooling experiment shown in Figure 3 are not surprising. The dropout or near drop out of certain strains is likely exacerbated by unequal cellular contributions from each of the overnight cultures, with slow growing mutants being added in lower quantities than fast or typical growing mutants. The authors should explicitly state that they recommend the original pooling protocol while providing advice for how the liquid pooling strategy could've been improved (endpoint OD600 measurements for all plates coupled with liquid contributions normalized to the individual mutant's optical density). I acknowledge that this last point would be incredibly tedious to accomplish manually, but inexpensive liquid handling robots are rapidly becoming available and accessible for many labs.

Response to reviewers

The referenced manuscript has been reviewed by two reviewers. Both have found the manuscript well written, with only a couple suggestions on potential improvements prior to publication. First, sequencing read data from Figure 3 should be available in a public repository. Second, reviewer 2 has provided two instances where quantitative quality control checkpoints could be added to the protocol. Please address these comments in a revision of the manuscript.

- The sequencing data has been deposited in NCBI GEO, and the accession number has been added to the manuscript (lines 244-246). We have addressed the concerns of reviewer 2 below.

Reviewer #1 (Comments for the Author):

In this manuscript, Willett and colleagues describe a protocol for generating an arrayed Tn library in *E. faecalis* that is generally applicable to other bacterial species. Arrayed Tn libraries are important resources for conducting gene function screens in bacteria, but they are time consuming and labor intensive to generate. As the authors also mention, there are no clear guidelines on how to generate such resources, with the 'secrets' of how to generate these libraries arising through a combination of intuition, luck and working in labs with long histories of making these resources. In this regard, it is helpful to have an established, detailed, protocol for creating an arrayed mutant library. Although the procedure is written for replication of the *E. faecalis* SmarT library it could be applied to other bacterial species with modifications (e.g. growth media). The method is rigorous and easy to follow, with the tables (1-3) being particularly helpful for following the procedure. Description of the reagents and equipment required is very detailed, and this will make the protocol easier to replicate. The goal of the manuscript was to provide a protocol for generation/replication of arrayed mutant libraries and the authors have successfully achieved this. I have no major or minor comments.

- We thank the reviewer for their comments and are pleased that others also see the utility in this protocol.

Reviewer #2 (Comments for the Author):

In this manuscript, Willett et al. develop a standardized protocol for the replication of strain libraries and apply it to copying a previously-created, transposon-mutagenized strain library of *Enterococcus faecalis* OG1RF. The authors use this protocol to create 15 copies of said library and discuss several quality control checkpoints, while exploring alternative strategies to pooled strain library creation. Given the increasing prevalence of high-throughput experiments within the field of microbiology, studies like this are needed and will proliferate concepts of industrial quality engineering. While the manuscript is well written and the experiments are straightforward, the reliance on qualitative observation for several of the quality control steps weakens the approach. The authors should implement a few simple changes to the protocol to improve the strength of the manuscript.

Major issues:

I agree with where the authors have placed quality control checkpoints within the library replication protocol but believe they could be made more rigorous by making these checks quantitative in nature.

1) Lines 120-125. The authors claim that because the mutant library was grown in deep well plates, and because deep well plates do not fit within standard plate readers, they did not measure optical density of overnight cultures. While the scale of the replication experiments is quite large, I find it difficult to believe that simply transferring a few hundred microliters of overnight culture to a standard 96-well plate and measuring OD600 would have been prohibitively difficult. The inclusion of an endpoint OD600 plate read for all library plates within the protocol would greatly improve the strength of the manuscript. By including this step, the authors would be able to better identify slow-growing contamination in blank wells and would have quantitative information on each mutant's endpoint growth --- valuable data to labs receiving the library and also data which could vastly improve liquid culture library pooling efforts.

- We agree that inclusion of an OD600 measurement would improve the protocol. We have not generated endpoint OD600 data at this time given that this would require another freeze-thaw cycle of the library. We have adjusted the manuscript (main text and supplementary protocol) to provide clear recommendations for other users to include this step (lines 118-120).

2) Lines 130-137. My criticism in point 1 holds for the dye leakage experiments. Similar, quantitative experiments using dyes with well-characterized absorbance spectra and plate readers are commonplace in liquid handling quality engineering contexts. While I don't doubt the results of their experiment (shown in Figure 2), the manuscript could be greatly improved by using quantitative data to demonstrate the lack of well-to-well leakage. This could be visualized as a heatmap.

- This is a good point, and we have updated Figure 2 and the corresponding manuscript text accordingly (lines 131-137, plus lines 221-231 in the methods section). Instead of using a dye, we set up a 96-well plate with *E. faecalis* OG1RF inoculated in a checkerboard pattern. We then sealed and mixed the plates following the exact protocol we used for the library plates. Plates were incubated at 37 °C overnight, after which A600 values were measured. Leakage between wells would result in OG1RF growth in the media blanks. We have provided this data in addition to the original dye experiment in the updated Figure 2.

Minor issues:

The results of the liquid pooling experiment shown in Figure 3 are not surprising. The dropout or near drop out of certain strains is likely exacerbated by unequal cellular contributions from each of the overnight cultures, with slow growing mutants being added in lower quantities than fast or typical growing mutants. The authors should explicitly state that they recommend the original pooling protocol while providing advice for how the liquid pooling strategy could've been improved (endpoint OD600 measurements for all plates coupled with liquid contributions normalized to the individual mutant's optical density). I acknowledge that this last point would

be incredibly tedious to accomplish manually, but inexpensive liquid handling robots are rapidly becoming available and accessible for many labs.

- We thank the reviewer for this point and have updated the manuscript accordingly (lines 172-174 and lines 193-198).

June 19, 2023

Dr. Julia Willett
University of Minnesota Twin Cities
Minneapolis, MN 55455

Re: Spectrum01693-23R1 (Optimized replication of arrayed bacterial mutant libraries increase access to biological resources)

Dear Dr. Julia Willett:

Your manuscript has been accepted, and I am forwarding it to the ASM Journals Department for publication. You will be notified when your proofs are ready to be viewed.

Sincerely,

Justin Kaspar
Editor, Microbiology Spectrum
